# A Novel Magnetic Nano-Sorbent Functionalized from N-methyl-D-glucamine for Boron Removal from Desalinated Seawater

**Tingting Sun** [1,2], **Fulin Li** [2,3,*], **Qikun Zhang** [4], **Xiaolin Geng** [5], **Huawei Chen** [2,3,*] and **Qi Zhao** [1,2]

1    School of Water Conservancy and Environment, University of Jinan, Jinan 250022, China;
     suntthnsq@163.com (T.S.); gstszq@163.com (Q.Z.)
2    Water Resource Research Institute of Shandong Province, Jinan 250014, China
3    Key Laboratory of Water Resources and Environment of Shandong Province, Jinan 250014, China
4    College of Chemistry, Chemical Engineering and Materials Science, Shandong Normal University,
     Jinan 250014, China; zhangqk@sdnu.edu.cn
5    Wudi County Water Conservancy Bureau of Shandong Province, Binzhou 251900, China;
     wdshuiziban@bz.shandong.cn
*    Correspondence: fulinli@126.com (F.L.); chenhuawei8036@163.com (H.C.)

**Abstract:** Boron is a special pollutant. Because of its relatively small molecular weight, it can freely penetrate the reverse osmosis membrane in the same way that water molecules can in reverse osmosis during seawater desalination, which affects the effluent quality of desalinated seawater. In this study, a new magnetic adsorption material, MNP-NMDG, was synthesized by combining magnetic nanoparticles (MNPs) of $Fe_3O_4$ with N-methyl-D-glucamine with a high selectivity to boron, and MNP-NMDG was characterized by scanning electron microscopy (SEM), Fourier transform infrared spectroscopy (FT-IR), and X-ray diffraction (XRD). The adsorption properties of the MNP-NMDG for boron during seawater desalination under static and dynamic conditions was studied from the aspects of pH, adsorbent dosage, adsorption kinetics, and isotherms. The results showed that according to the breakthrough curve of dynamic adsorption, MNP-NMDG had a high boron-adsorption capacity, and the static adsorption capacity was 9.21 mg/g. The adsorption performance was the best at pH = 9, and the adsorption equilibrium was achieved within 40 min. Boron adsorption conformed to the Freundlich adsorption isotherm and to the pseudo-second-order kinetic model. This composite material not only provides an effective and rapid way to remove boron from desalinated seawater, but also has a shorter removal time and makes it more easily separated using the external magnetic field.

**Keywords:** desalinated seawater; boron; N-methyl-D-glucamine; magnetic separation; adsorbent

## 1. Introduction

   Water resources account for about three-quarters of the earth's surface, but there is a serious shortage of drinking water in many countries around the world [1,2], mainly in developing countries and countries in the Middle East [3]. The reason for this situation is that seawater and bitter brine account for 97.5% of the total water resources on the Earth, while freshwater accounts for only 2.5% of the total water resources on the Earth, of which 80% are distributed in the form of ice in the Arctic and in Antarctica, and on mountains [4,5]. In the face of increasing water demands and a diminishing freshwater supply [6], desalination has become the key to helping meet the growing demand for water [7,8], especially in water-deficient countries, where the supply of desalinated seawater far exceeds the supply of natural freshwater [9]. With changes in the sea area and season, seawater usually contains 4–6 mg/L boron, and there is still a small amount of boron that exceeds the recommended standard in desalinated seawater. Boron plays an irreplaceable role in the growth and development of humans and animals [10]. However, excessive boron intake can cause a series of adverse reactions and even boron poisoning. Among seawater

desalination technologies, reverse osmosis (RO) is becoming more and more popular because of its convenient operation and low cost. In China, 68.7% of seawater desalination projects adopt reverse osmosis (RO). Although the desalination rate of seawater desalination processes has reached more than 99%, the removal of boron ions is not ideal [11]. The boron removal rate of reverse osmosis in conventional seawater is only 60–80% [12]. The boron concentration in the produced water seriously exceeds standards, and the boron content in desalinated seawater is 0.5–2.5 mg/L [13], which exceeds the amount of 0.5 mg/L that has been specified in the hygienic standards for drinking water (GB5749–2006) [14]. In 2017 and after several revisions, the World Health Organization set the limit for boron in drinking water to 2.4 mg/L [15]. Many countries and organizations have also developed corresponding standards. In the United States, the boron content limits in drinking water are generally 0.5 mg/L–0.9 mg/L [16]; in California, Britain, Israel, and Japan, the limit is 1.0 mg/L [17–19]; in New Zealand and New Jersey, the limit is 1.4 mg/L [20]; and the upper recommended limit for European drinking water is 1.5 mg/L [21]. Saudi Arabia follows the WHO drinking water standard, and Australia and Canada have higher boron content limits for drinking water, 4.0 mg/L and 5.0 mg/L, respectively [22,23]. Although various countries are different in terms of the limits of boron content in desalinated seawater, boron removal technology for seawater desalination is still an essential part of the whole desalination process.

At present, many studies have discussed how to remove high levels of boron from wastewater, including adsorption methods, chemical precipitation, the reverse osmosis method [24], extraction methods, and electrocoagulation methods [25]. Yet, most conventional boron removal methods do not meet the process requirements for seawater desalination. The main methods for boron removal during seawater desalination are electrodialysis [26], ion exchange resins [27], as well as the combination of various methods for desalination and boron removal [28]. So far, the most effective adsorbent to remove boron from an aqueous medium is boron-specific resin, which usually has a N-methyl-D-glucamine functional group (NMDG) [29]. Hydroxyl functional groups (NMDG) can form selective complexes with boron in the presence of many ions to achieve boron removal. However, NMDG has a small molecular weight and is easily soluble in water, so it is necessary to develop insoluble NMDG derivatives. In this study, a new magnetic adsorption material was synthesized successfully by combining $Fe_3O_4$ magnetic nanoparticles (MNPs) with N-methyl-D-glucamine (NMDG) with high selectivity to boron. $Fe_3O_4$ nanoparticles were insoluble in water and had magnetism, which was convenient in regard to separation and recovery from water. Hydroxyl functional groups (NMDG) can selectively complex with boron in the presence of many ions to achieve the purpose of removing boron. The preparation method for this adsorbent includes three steps: the preparation of magnetic nanoparticles (MNPs) of $Fe_3O_4$, the encapsulation of the magnetic nanoparticles in $SiO_2$, and the creation of the composite with N-methylglucosamine. This new adsorbent is suitable for removing boron from desalinated seawater. Compared to the traditional boron-selective adsorbent, the combination of functional groups and magnetic nanoparticles is not only suitable for water with a low boron concentration, but it is also convenient for recycling.

## 2. Materials and Methods

### 2.1. Experimental Materials

Iron trichloride hexahydrate, iron sulfate heptahydrate, sodium metasilicate nonahydrate, and sodium hydroxide were purchased from Sinopharm Chemical Reagent Co., Ltd. (Shanghai, China). Nitric acid (65–68%) was provided by Laiyang Kangde Chemical Co., Ltd. (Laiyang, China). N-methyl-D-glucamine (99%), N-Hydroxysuccinimide (NHS), and N-(3-Dimethylaminopropyl)-N′-ethylcarbodiimide hydrochloride (EDC) were provided by Shanghai Macklin Biochemical Technology Co., Ltd. (Shanghai, China). All of the reactants were of analytical grade.

### 2.2. *Synthesis of Sorbent*
### 2.2.1. Synthesis of Fe$_3$O$_4$ Nanoparticles

The Fe$_3$O$_4$ nanoparticles were prepared by means of the co-precipitation method. The specific steps were as follows: an amount of 1.623 g FeCl$_3$·6H$_2$O and 1.082 g FeSO$_4$·7H$_2$O were dissolved in deionized water to prepare a solution containing 0.30 mol/L iron salt (the concentration ratio of Fe$^{2+}$ to Fe$^{3+}$ is 1.00:1.50), which was placed in a 250 mL flask. The solution was heated to 60 °C in a water bath, stirred, and 0.25 mol/L NaOH solution was added dropwise. The mixed solution gradually changed from orange to black, and a large number of black granular solids were generated. When the pH increased to 9–10, the mixture was stirred for 30 min. After being allowed to age for a certain time, mixing was stopped, and then a magnet was put under the container. The mixture was kept still until the black powder was fully immersed at the bottom of the container. The solution became almost transparent, and the supernatant was removed using a liquid transfer tube. The black particles were washed three times with deionized water and dried in vacuum to obtain Fe$_3$O$_4$ nanoparticles.

### 2.2.2. SiO$_2$ Wrapped Magnetic Fe$_3$O$_4$ Nanoparticles

A 0.5 g amount of Fe$_3$O$_4$ nanoparticles was placed in a 250 mL triple-mouth flask with 0.6 g Na$_2$SiO$_3$·9H$_2$O and 100 mL of deionized water. The mixture experienced strong electric stirring, was heated to 60 °C in a stirred water bath, and the 0.25 mol/L nitric acid solution was added dropwise. The pH of the system was adjusted to about 6, and the mixture was heated for about 60 min in a water bath, and then stirring stopped. Finally, a magnet was placed under the flask, and when the black powder was immersed at the bottom of the flask, and the upper liquid was removed. The black powder was centrifuged with deionized water and washed three times, and the final product was dried in a vacuum-drying oven at 70 °C for 12 h. After drying, a nano-sized iron oxide wrapped in SiO$_2$ was obtained, and the product was sealed in a weighing flask.

### 2.2.3. Synthesis of MNP-NMDG

A 0.2 g amount of SiO$_2$-modified magnetic nano-Fe$_3$O$_4$ particles was weighed and dispersed in 10 mL deionized water, and then 1 mL of EDC solution and NHS solution were added to the suspension and ultrasonically dispersed for 5 min. The MNP-NMDG was prepared by adding 0.3 g N-methyl-D-glucamine, shaking the prepared reagent in a constant temperature oscillation box for 2 h, putting the magnetic iron under the bottle to separate and remove the mother liquor, followed by centrifugal washing with deionized water three times and drying at 50 °C for 12 h in a drying closet (Figure 1).

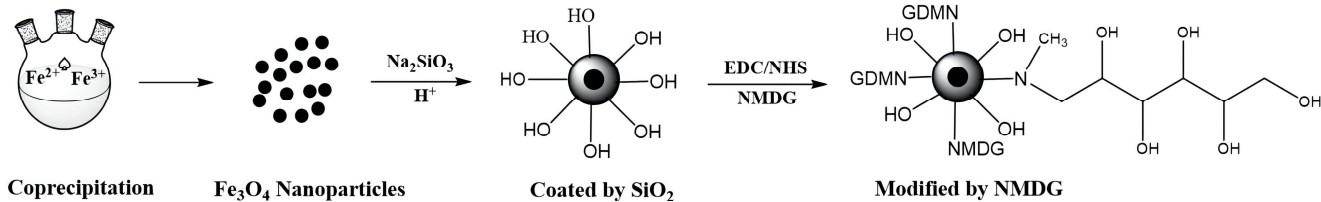

**Figure 1.** Sketch for the preparation of magnetic absorbent MNP-NMDG.

### 2.3. *Characterization Experiments of Fe$_3$O$_4$ and Its Magnetic Composites*

Scanning electron microscopy (SEM), Fourier transform infrared spectroscopy (FT-IR), and X-ray diffraction (XRD) were used to characterize the appearance and chemical composition of the adsorbent. Scanning electron microscopy (SEM) was used to analyze the surface morphology of the composites using a Hitachi high-tech field emission scanning electron microscope SU8200 (Hitachi High-Tech Group, Tokyo, Japan). The FT-IR spectra of the Fe$_3$O$_4$ nanoparticles and MNP-NMDG were analyzed using a Nicolet/Nexus 670 infrared spectrometer (Nicolet, Madison, WI, USA), and the FT-IR spectrum was obtained with a resolution of 4 cm$^{-1}$ and at a scanning range of 400–4000 cm$^{-1}$. The composite

materials were analyzed by XRD in the range of 10–80° using a Rigaku D/Max-2400 X-ray diffractometer (Rigaku Corp, Tokyo, Japan).

*2.4. Adsorption Experiments of MNP-NMDG Magnetic Composites*

Boron was removed from desalinated seawater by batch experiments and flow-through column experiments. In the batch experiments, the effects of the adsorption time, adsorbent dosage, pH, and initial concentration of boron on the adsorbent were studied, and the adsorption effect of the boron absorbent was studied using adsorption isotherm and kinetic methods. Different concentrations of a boric acid solution were configured, and an appropriate amount of adsorbent was mixed with it in a conical flask; then, the conical flask containing the mixture was placed in a constant temperature oscillator to oscillate. Finally, the supernatant was collected by magnetic separation, and the absorbance was measured using an ultraviolet spectrophotometer and compared with the standard curve to determine the boron concentration in the adsorbed water.

For the flow-through column experiments, a flow-through column device was established using a peristaltic pump and an acid-type titration tube. As shown in Figure 2, a certain amount of the prepared magnetic adsorption material, MNP-NMDG, is put into the acid-type titration tube. At room temperature, desalinated seawater is input from the upper end of the burette by the peristaltic pump at a certain inlet velocity that was pre-determined for the boron removal experiments. Samples were taken from the outflow at different intervals, and the number of boron ions in the raw desalinated seawater and in the water after the adsorption treatment was detected by methimino-H spectrophotometry. The breakthrough curve was plotted, and the dynamic boron removal adsorption capacity was calculated. The adsorption efficiency and the removed boron rate was computed following Equations (1)–(3):

$$q_e = \frac{(C_0 - C_e)V}{m} \tag{1}$$

$$q_t = \frac{(C_0 - C_t)V}{m} \tag{2}$$

$$\%Removal = \frac{(C_0 - C_e)}{C_0} \times 100 \tag{3}$$

where $C_0$ (mg·L$^{-1}$) is the initial boron concentration, while $C_e$ and $C_t$ (mg·L$^{-1}$) are the equilibrium and the boron concentration at time, $m$ (g) is the weight of the sorbent, $V$ (L) is the volume of the aqueous solution, and $q_e$ and $q_t$ (mg·g$^{-1}$) are the amount of boron adsorbed at equilibrium and at any time, respectively.

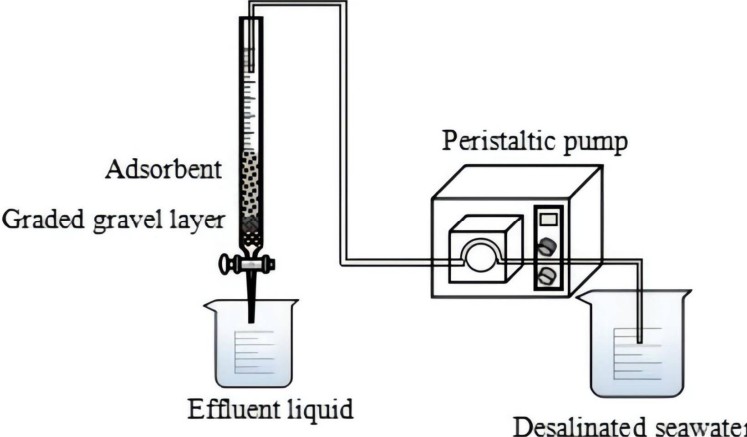

**Figure 2.** Experimental device for dynamic adsorption to remove boron from desalinated seawater using MNP-NMDG.

## 3. Results and Discussion

### 3.1. Characterization of Fe$_3$O$_4$ and Its Magnetic Composites

Figure 3 shows High Resolution Transmission Electron Microscope (HRTEM) images of bare (Figure 3a) and silica coated Fe$_3$O$_4$ nanoparticles (Figure 3b), the final magnetic absorbent (Figure 3c), and an SEM image of the magnetic nanocomposite MNP-NMDG (Figure 3d,e). From the SEM results in Figure 3d, it can be seen that MNP-NMDG is spherical in shape and has a relatively uniform and small particle size. From the SEM results in Figure 3e, it can be seen that the surface of the material is uneven and that there are tiny bumps. Because magnetic nanoparticles of Fe$_3$O$_4$ and SiO$_2$-coated particles are spherical (Figure 3a,b), this can explain the adhesion of N-methyl-D-glucamine on the outer surface. MNP-NMDG has a morphology in which N-methyl-D-glucamine is attached to the surface of the magnetic material Fe$_3$O$_4$@SiO$_2$, which is conducive to boric acid adsorption.

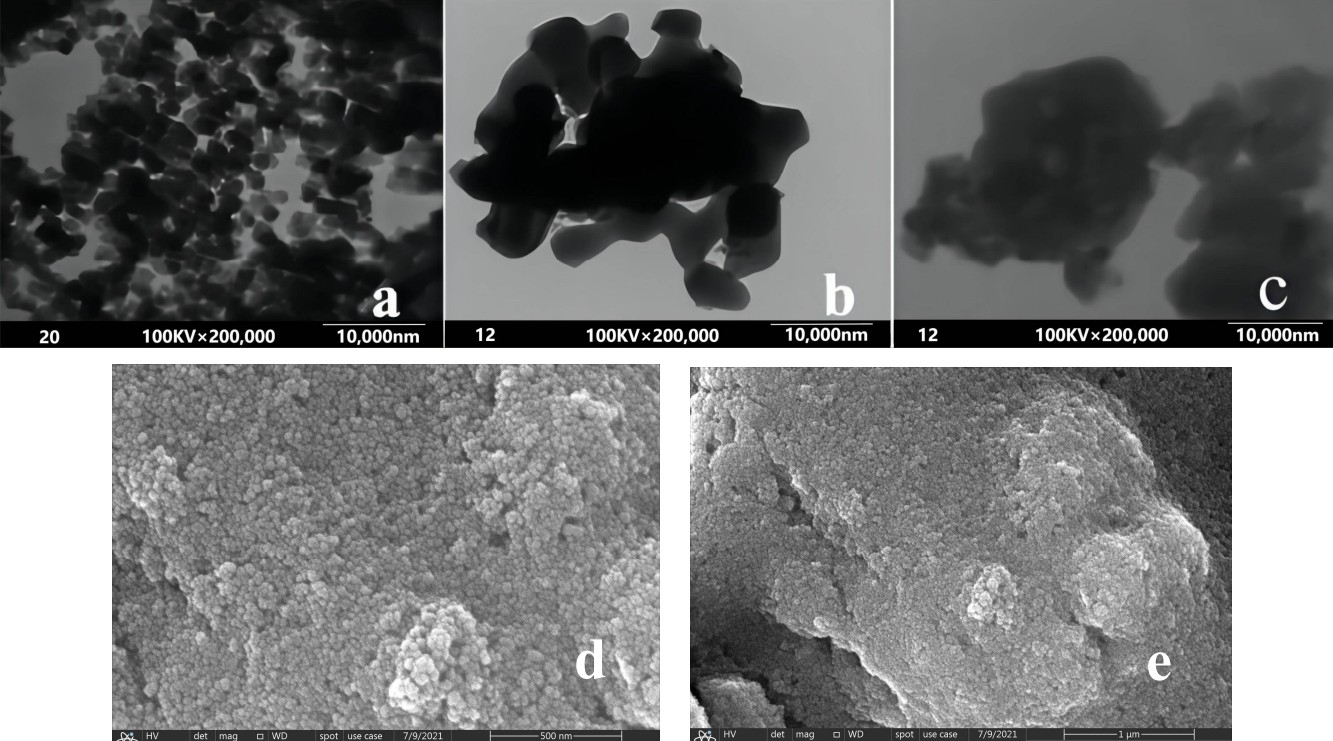

**Figure 3.** HRTEM images of bare (**a**) and silica coated Fe$_3$O$_4$ nanoparticles (**b**), the final magnetic absorbent (**c**), and an SEM image of MNP-NMDG (**d**,**e**).

The Fourier transform infrared (FT-IR) spectra of Fe$_3$O$_4$ and MNP-NMDG are shown in Figure 4. In the figure, the absorption peak at 567 cm$^{-1}$ corresponds to the vibration of the Fe-O bonds in the crystalline lattice of Fe$_3$O$_4$ [30]. Compared with the bare magnetic nanoparticles, new bands were present when the magnetic Fe$_3$O$_4$ nanoparticles were coated with silica. The absorption peak at 3450 cm$^{-1}$ corresponds to the stretching vibration of the O-H bonds in the NMDG structure. The absorption peak at 1650 cm$^{-1}$ corresponds to the stretching vibration of C-O, while the absorption peak at 1100 cm$^{-1}$ corresponds to the asymmetric stretching vibration absorption peak of Si-O-Si (Figure 4b) [31]. Figure 4c shows the infrared spectrum obtained for the final magnetic absorbent. The figure shows that the intensity of the absorption peak at 3450 cm$^{-1}$ increases significantly, as a result of the stretching vibration of -O-H [32], and indicates that NMDG was immobilized successfully on the magnetic nanoparticles.

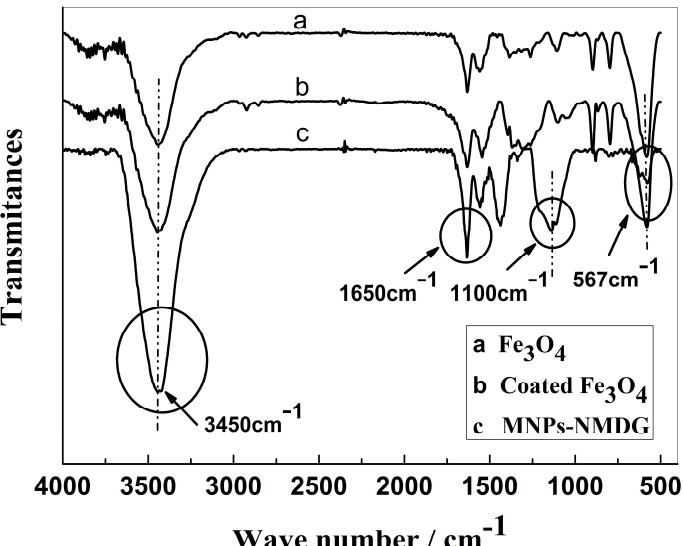

**Figure 4.** FT-IR spectra of bare (**a**) and coated $Fe_3O_4$ nanoparticles (**b**), and the final magnetic absorbent (**c**).

In order to determine the formation of the MNP-NMDG composites, powder X-ray diffraction (XRD) analysis was carried out on $Fe_3O_4$, $Fe_3O_4@SiO_2$, and MNP-NMDG. As shown in Figure 5, six peaks could be clearly observed, and the corresponding diffraction angles were 30.29°, 35.67°, 43.29°, 53.66°, 57.17°, and 62.86°, which belonged to the (220), (311), (400), (422), (511), and (440) crystal planes of $Fe_3O_4$ powder [33]. The diffraction patterns of $Fe_3O_4@SiO_2$ and MNP-NMDG were similar to those of $Fe_3O_4$, and there were no obvious differences in the positions of the diffraction peaks. $Fe_3O_4@SiO_2$ has an intensity that is relatively weaker, which may be due to the $SiO_2$ coating on the surface of the $Fe_3O_4$ nanoparticles, and it was also demonstrated that $SiO_2$ wrapping has no effect on the structure of $Fe_3O_4$ nanoparticles. The strength of MNP-NMDG is weaker than both, indicating that $Fe_3O_4@SiO_2$ is the main component of the MNP-NMDG composites and that it helps to maintain a good crystal structure. At the same time, the introduction of N-methyl-D-glucamine did not interfere with the structure of $Fe_3O_4@SiO_2$. The MNP-NMDG composites did not contain the peak value of N-methyl-D-glucamine, which is likely due to the relatively low content of N-methyl-D-glucamine in the composite and its high dispersion in the aqueous solvent.

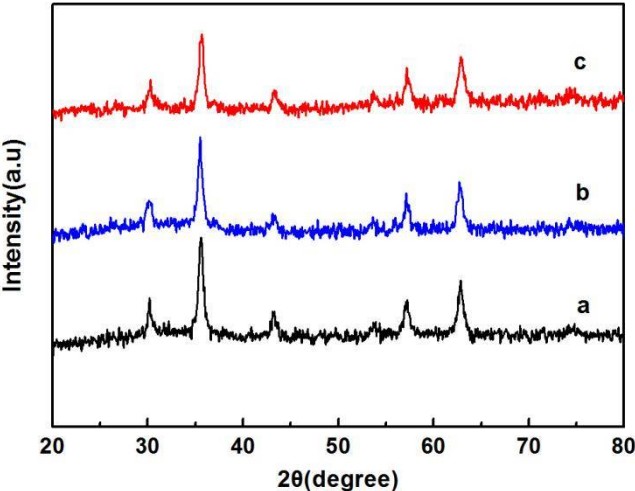

**Figure 5.** Powder X-ray diffraction pattern of $Fe_3O_4$ (**a**), $Fe_3O_4@SiO_2$ (**b**), and MNP-NMDG (**c**).

*3.2. Boron Adsorption Test for MNP-NMDG*

3.2.1. Effect of Solution pH on Boron Removal by MNP-NMDG

The effects of different pH values on boron adsorption by MNP-NMDG were studied at 25 °C. Eight conical flasks were used, and a 25 mL volume of boric acid solution was added into the flasks to ensure that the concentration of boric acid in each group was 1 mg/L along with 40 mg/L of the adsorbent. The pH was adjusted using 0.1 mol/L of a diluted nitric acid solution and a sodium hydroxide solution by means of a micro-sampler. The conical flask was placed in a water bath with a constant temperature oscillator set at 25 °C for 40 min to be tested. The pH value affects the adsorption process by changing the existing form of boron. The boron adsorption experiment was carried out by adjusting the solution's pH to be between 3 and 10. As shown in Figure 6, the adsorption capacity of the composite material for boron first decreases and then increases, and then decreases as the pH increases from 3 to 10. The adsorption capacity was the largest at pH = 9, followed by pH = 3, which indicated that the adsorbent has a good boron adsorption effect under acid and alkali conditions. At pH = 3, the main form of boric acid is $H_3BO_3$, and at pH = 9, the main form of boric acid is $B(OH)_4^-$. $B(OH)_4^-$ forms a stable complex with the hydroxyl structure of NMDG (Figure 7), which is also the reason why the adsorption capacity is larger at pH = 9. However, considering that the water quality should meet the standards for drinking water, pH = 7 was selected as the adsorption condition.

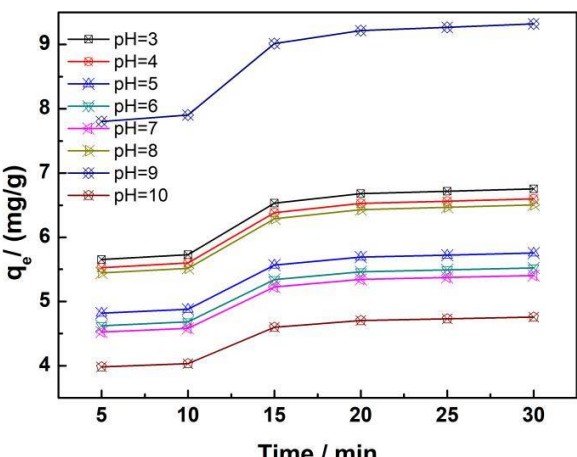

**Figure 6.** Effect of the solution pH on boron removal by MNP-NMDG (the concentration of boric acid was 1 mg/L, 25 mL volume of boric solution, along with 40 mg/L of the adsorbent, and the oscillation time was set to 40 min).

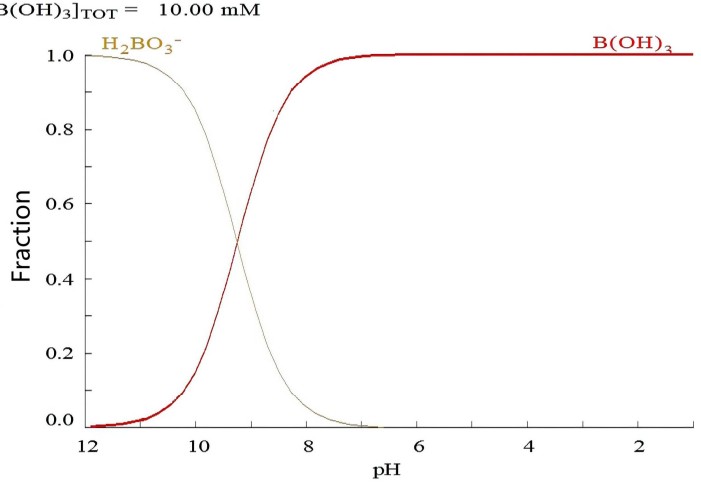

**Figure 7.** The diagram related to the effect of pH on boric acid and borate distribution.

### 3.2.2. Effect of Adsorbent Dose on Boron Removal by MNP-NMDG

The amount of adsorbent directly affects its adsorption capacity. In order to obtain the best adsorption conditions, 0.5–3.0 mg of composite materials (with doses of 20–120 mg/L) was added to a series of 25 mL volume and 1 mg/L concentration boric acid solutions to carry out different experiments (GE Electronic Balance, Shanghai Youke Instrument Co., Ltd. (Shanghai, China)), and the oscillation time was 40 min. As shown in Figure 8, the boron removal rate by composite materials shows an obvious increase as the dosage of the adsorbent increases, but the increasing trend slows down when the dosage exceeds 2 mg. Considering the adsorption effect and cost, a dosage of 1 mg was selected for the following adsorption experiments. It can be seen from the curve in the figure that the adsorption capacity ($q_e$) decreased as the adsorbent dosage increased. This is because the active sites on the adsorbent surface were not saturated with boron. When the boron concentration is constant, the increase in the dosage of the adsorbent will lead to a decrease in the boron concentration compared to the amount of the adsorbent in a water-based medium. Therefore, the adsorbent is in contact with less boron ions, which means that the active sites in the adsorbent are unsaturated.

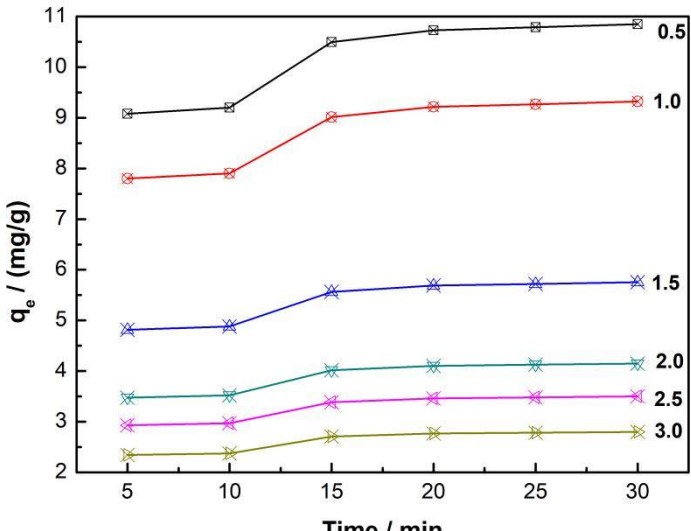

**Figure 8.** Effect of adsorbent dose on boron removal by MNP-NMDG (the concentration of boric acid was 1 mg/L, 25 mL volume of boric solution, pH = 7, and the oscillation time was set to 40 min).

### 3.2.3. Adsorption Kinetics

The effect of the adsorption time on the adsorption of boron by MNP-NMDG was studied at 25 °C. Six boric acid solutions with 25 mL volume and 1 mg/L concentration were prepared, and 1 mg of adsorbent (with a dose of 40 mg/L) was placed in a conical flask and mixed with it; then, the conical flask containing the mixture was put into a constant temperature oscillator and oscillated for 5–30 min. The boron concentration in the solution decreased rapidly with the prolongation of the adsorption time (Figures 6 and 8). The speed gradually slowed down within 15–25 min, and after 25 min, the boron concentration increased slightly, which may be due to the slight desorption of the adsorbed substance, but the concentration did not change after 40 min, indicating that the adsorption process had reached equilibrium under the experimental conditions. In order to ensure complete adsorption, the static adsorption time was 30 min in the following experiments.

In order to describe the kinetic process of MNP-NMDG on boron more clearly, taking 1 mg/L of boron solution and a 2 mg dose MNP-NMDG, the pseudo-first-order kinetic equation and pseudo-second-order kinetic equation were used to show the fit at 25 °C [34].

The pseudo-first-order dynamic equation and pseudo-second-order dynamic equation are shown in Equations (4) and (5) [35]:

$$\ln(q_e - q_t) = \ln q_e - \frac{k_1}{2.303}t \tag{4}$$

$$\frac{t}{q_t} = \frac{1}{k_2 q_e^2} + \frac{t}{q_e} \tag{5}$$

where $k_1$ (min$^{-1}$) and $k_2$ (g·mg$^{-1}$·min$^{-1}$) are the pseudo-first-order and pseudo-second-order adsorption rate constants, respectively, $q_e$ and $q_t$ (mg·g$^{-1}$) are the amount of adsorbed boron at adsorption equilibrium and at any time, and $t$ (min) is the adsorption time.

The fitted kinetic data from the model are shown in Figure 9 and in Table 1. The $R^2$ values in the pseudo-first-order kinetic model and in the pseudo-second-order kinetic model of MNP-NMDG are 0.8933 and 0.9981, respectively. The adsorption capacity of the pseudo-first-order kinetics at adsorption equilibrium was 4.48 mg/g, which is quite different from the equilibrium adsorption capacity of 9.21 mg/g in the experiment. The adsorption equilibrium amount of the pseudo-second-order kinetics at adsorption equilibrium was 9.59 mg/g, which is close to the adsorption capacity at the actual adsorption equilibrium. This indicates that the boron adsorption by MNP-NMDG conforms to the pseudo-second-order kinetic model, that is, the adsorption rate is controlled by chemical adsorption and involves electronic covalent or electronic migration between the adsorbate and adsorbent [36].

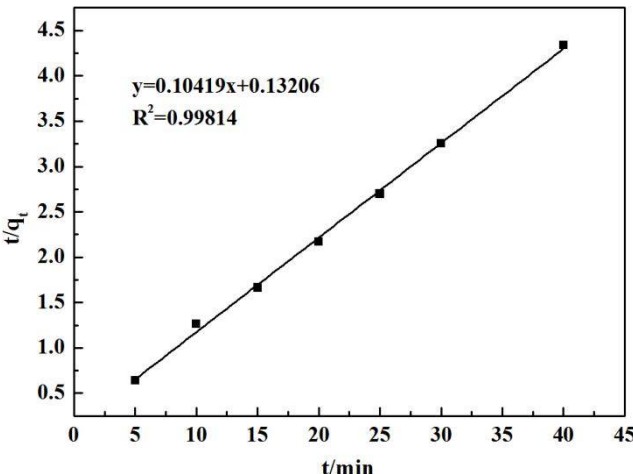

**Figure 9.** Linear fit curve of a quasi-secondary kinetic model for MNP-NMDG.

**Table 1.** Kinetic parameters of MNP-NMDG for boron adsorption.

| Model | $k$ | $R^2$ | $q_e$ (mg·g$^{-1}$) |
|---|---|---|---|
| the pseudo-first-order | $k_1 = 0.41$ | 0.8933 | 4.48 |
| the pseudo-second-order | $k_2 = 0.08$ | 0.9981 | 9.59 |

### 3.2.4. Adsorption Isotherm

Six 25 mL volumes of boric solution with different concentrations were configured, and 0.1 mol/L of diluted nitric acid or sodium hydroxide solution was used to adjust the pH value of the solution to 7. Then, 1 mg of adsorbent (with a dose of 40 mg/L) was mixed with it in a conical flask. The conical flask containing the mixture was placed in a constant temperature oscillator for 40 min. It can be seen from Figure 10 that with the increase in the concentration of the boron solution from 1 mg/L to 25 mg/L, the adsorption capacity of the composite material for boron also increases continuously.

The Freundlich model, which is an empirical equation based on the adsorption phenomenon on a uniform surface, was used to analyze the boron adsorption by MNP-NMDG. The corresponding parameters are shown in Table 2, and the mathematical linear formula is shown in Equation (6) [37]:

$$\ln q_e = \ln K_F + \frac{1}{n} \ln C_e \tag{6}$$

where $C_e$ (mg·L$^{-1}$) is the boron concentration at adsorption equilibrium, $q_e$ (mg·g$^{-1}$) is the amount of adsorbed boron at adsorption equilibrium, $K_F$ (L·mg$^{-1}$) is the Freundlich adsorption coefficient, and $n$ is the Freundlich constant.

According to Freundlich model fitting, the parameter n can indicate adsorption difficulties. In this experiment, $1/n$ is 0.14 (Table 2), which is between 0.1 and 0.5, indicating that the boron in the solution was easily adsorbed by MNP-NMDG.

**Table 2.** Freundlich adsorption isotherm fitting parameters of MNP-NMDG for boron adsorption.

| Elements | $1/n$ | $K_F$ | $R^2$ |
|----------|-------|-------|-------|
| B | 0.14 | 6.02 | 0.8748 |

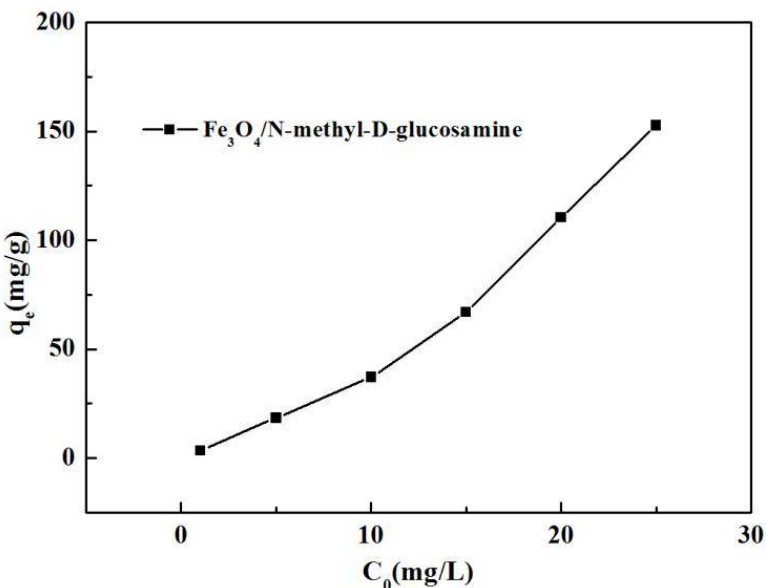

**Figure 10.** Effect of original concentration on boron removal by MNP-NMDG (pH = 7, 25 mL volume of boric solution along with 40 mg/L of the adsorbent, and the oscillation time was set to 40 min).

### 3.2.5. Flow-Through Column Experiments

The breakthrough curve of MNP-NMDG is shown in Figure 11 and was obtained under the following conditions: room temperature, a flow rate of 3 mL/min, a 1.5 mass of MNP-NMDG in the column, and desalinated seawater with a boron concentration of 2.72 mg/L. In the curve, t is the time required for the flow-through column experiments, and C and $C_0$ are the boron concentrations of the effluent and feed, respectively. With the injection of desalinated seawater, the composite adsorption material gradually became fully saturated. When the inlet flow rate was 3 mL/min, boron leakage occurred in about 20 min, and a significant jump occurred at 20–60 min until penetration. The saturation time of the composite adsorption material was between 80–100 min. Generally, this remarkable leap indicates that the composite material has a high selectivity for boron ions in desalinated seawater [38].

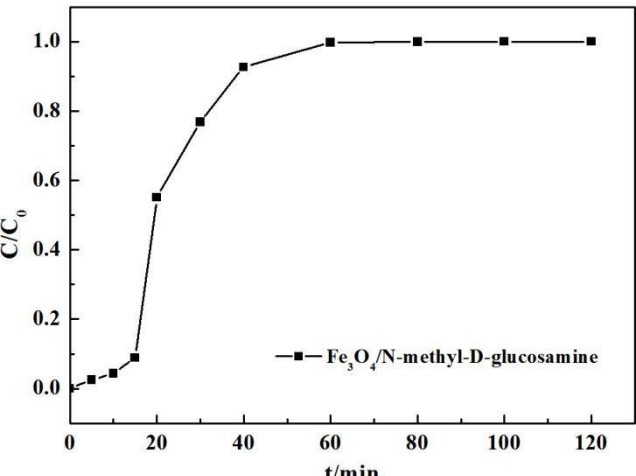

**Figure 11.** Effect of adsorption time on boron removal by MNP-NMDG (room temperature, a flow rate of 3 mL/min, a 1.5 mass of MNP-NMDG in the column, and boron concentration of 2.72 mg/L).

### 3.3. Mechanism of Boron Removal

In a water environment (neutral pH), boron mainly exists in the form of boric acid. Because of its high solubility, it is difficult to remove boron from the desalination membrane during traditional water treatment process. It was previously found that the N-methyl-D-glucosamine (NMDG) functional group is a polyol with five hydroxyl groups and one triamine terminal that can provide more complexing sites and that has high boron selectivity. Under acidic conditions, boron mainly exists in the form of $B(OH)_3$ acid. $B(OH)_3$ reacts with the neighboring polyol groups (Figure 12), and it will release protons and move to the left to achieve equilibrium, which explains the reason for the low boron recovery rate at low pH levels [37]. Under alkaline conditions, it exists in the form of $B(OH)_4^-$ (Equation (7)), and $B(OH)_4^-$ is complexed with NMDG to form a complex (Figure 13) [39]. In the process, hydrogen ions are released and neutralized by amine groups (Equation (8)). When the free hydrogen ions are neutralized by amine groups, the amount of boric acid in the aqueous solution moves to the right, thus promoting the continued complexation of $B(OH)_4^-$ with NMDG. The experiments were carried out in a weak alkaline environment because the pKa of boric acid in water is 9.05, and the proportion of $B(OH)_4^-$ in solution increases in alkaline conditions, so the optimum pH value for boron removal by boron specific resin is around 9.

**Figure 12.** Complexation reaction under acidic conditions.

**Figure 13.** Complexation reaction under alkaline conditions.

Boric acid solution:

$$B(OH)_3 + H_2O \rightarrow B(OH)_4{}^- + H^+ \tag{7}$$

$$-CH_2-N(CH_3)-CH_2- + H^+ \rightarrow -CH_2-N^+H(CH_3)-CH_2- \tag{8}$$

## 4. Conclusions

In the present work, $Fe_3O_4$ nanoparticles were synthesized by means of the coprecipitation method and compounded with N-methyl-D-glucamine with a high selectivity for boron to prepare a new magnetic adsorption material, MNP-NMDG. The MNP-NMDG was characterized by scanning electron microscopy (SEM), Fourier transform infrared spectroscopy (FT-IR), X-ray diffraction (XRD), and other technical means, and the boron removal ability and adsorption characteristics in desalinated seawater in static and dynamic states were studied. According to the breakthrough curve for dynamic adsorption, it can be seen that MNP-NMDG has a high boron adsorption capacity, and the static adsorption capacity is 9.21 mg/g. The adsorption performance was the best at pH = 9, and adsorption equilibrium was reached within 40 min. The boron adsorption observed here is in accordance with the Freundlich adsorption isotherm and the pseudo-second-order kinetic model, which indicates that adsorption is mainly influenced by chemical reactions. The prepared material has both ferromagnetism and boron removal specificity, is convenient to prepare and recover, is suitable for boron removal process in systems with a low boron content, and avoids the environmental pollution problems that are associated with traditional adsorbent preparation methods. This material has potential application prospects in boron removal from desalinated seawater.

**Author Contributions:** Conceptualization, F.L., H.C., Q.Z. (Qikun Zhang) and T.S.; methodology, F.L., H.C., Q.Z. (Qikun Zhang), X.G., T.S. and Q.Z. (Qi Zhao); software, T.S.; validation, F.L., H.C., Q.Z. (Qikun Zhang) and T.S.; formal analysis, T.S.; investigation, T.S.; data curation, T.S.; writing—original draft preparation, T.S.; writing—review and editing, F.L., H.C., Q.Z. (Qikun Zhang), X.G., T.S. and Q.Z. (Qi Zhao); project administration, F.L. and H.C.; funding acquisition, F.L. and H.C. All authors have read and agreed to the published version of the manuscript.

**Funding:** This research was financially supported by the National Key Research and Development Program of China (2018YFC0408006, 2021YFC3200504), the Key Hydraulic Engineering Research and Experiment Project for River Basin Water Conservancy Management and Service Center of Shandong Province (XQHFHZL-KY202004), the Ministry of Water Resources Science and Technology Promotion Project (SF-201803), the Ministry of Water Resources' Special Funds for Scientific Research on Public Welfare (201401003), the International Science and Technology Cooperation Program of China (2012DFG22140), and the sub-project of the National Science and Technology Major Project on Water Pollution Control and Treatment (2012ZX07404-003).

**Institutional Review Board Statement:** Not applicable.

**Informed Consent Statement:** Not applicable.

**Data Availability Statement:** The data used to support the findings of this study are included within the article.

**Acknowledgments:** The authors acknowledge valuable comments from the reviewers, which led to significant improvement of the paper.

**Conflicts of Interest:** The authors declare no conflict of interest.

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
