# Peer review of "A Novel Magnetic Nano-Sorbent Functionalized from N-methyl-d-glucamine for Boron Removal from Desalinated Seawater"

_water, doi:10.3390/w14081212_

Round 1

Reviewer 1 Report

Reviewer comments to the Authors:

COMMENTS FOR AUTHORS

Manuscript ID: water-1655800

Title:A Novel Magnetic Nano-sorbent Functionalized of N-methyl-D- 2glucamine for Boron Removal from Desalinated Seawater

As reviewer of your paper I have to thank you for submitting this paper for Water, but regret to advise you that it is not acceptable for publication in its present form, and will require revision (major) before it could be considered for publication.

I would be happy to consider a rewritten and resubmitted paper. In revising your paper, you should consider all the points. When resubmitting your paper, you must provide a point by point list of your response to the reviewer.

Below are major comments:

  1. Materials and Methods
  • Line 87 Iron ulfate heptahydrate correct to Iron sulfate heptahydrate
  • How the experiments were conducted (with how many repetitions for the examined parameters)? Please specify

Move text line

  • Move text line (213-219) in the part 2. Materials and Methods

3.2.1. Effect of Solution pH on Boron Removal by MNPs-NMDG

  • Provide some diagram related to the Effect of pH on boric acid and borate distribution. You can also create it in hydra/medusa. Hydra/medusa - chemical equilibrium database and plotting software (KTH Royal Institute of Technology). Hydra contains a database with logK data at 25°C. Run this program to define the chemical system you are interested on

https://www.kth.se/che/medusa

Author Response

Dear Reviewer,

Yours Sincerely,

Tingting Sun

Reviewer 2 Report

This manuscript brings results of synthesis of new magnetic functionalized composite and its applicability as adsorbent for Bo removal from aqueous systems. The manuscript presents an important area of research, and the findings are promising; however, there are several points which should be addressed before the manuscript is accepted for publication:

Lines 23-24: ” and the static adsorption capacity was 9.2171 mg/g”

9.21 instead of 9.2171.

Lines 108-109: ”Fe3O4 nanoparticles of 0.5 g and 0.6 g Na2SiO3·9H2O were dissolved in 100 mL deionized water”

Fe3O4 nanoparticles were just added in 100 mL deionized water. I do not believe they were dissolved. This must be rewritten.

Line 139: ”Boron in desalinated seawater was removed by static and dynamic methods”

Both ”static method” and ”dynamic method” are rather confusing terms. By ”static method” it could be understood experiments without any kind of mixing (non-disturbed). The ”dynamic method” term is too vague. You should replace throughout the manuscript ”static method” with ”batch experiments”, and ”dynamic method” with ”flow-through column experiments”

Line 212: ” 3.2.1. Effect of Solution pH on Boron Removal”.

You must provide at this section the following experimental parameters:

(1) volume of boric solution added in the conical flask,

(2) the dose of adsorbent, in mass/L,

(3) mixing (oscillation) intensity.

I strongly recommend replacing Fig.6 by a figure containing 8 curves (one curve for each pH) depicting the decrease of Bo concentration with time.

Line 234: ” 3.2.2. Effect of Adsorbent dose on Boron Removal”.

You must provide at this section the following experimental parameters:

(1) volume of boric solution added in the conical flask,

(2) dose of adsorbent, in mass/L.  Are you sure that the added masses (0.5-3) are milligrams and not grams? Please double check! If you confirm that milligram is correct, then please give in the manuscript the name and manufacturer of the balance used to weigh 0,0005 g.

(3) mixing (oscillation) intensity.

Lines 237-239: ”As shown in Figure 7, the removal rate of boron by composite materials increases obviously with the increase of adsorbent dosage, and the increase trend will slow down when the dosage exceeds 2 mg”.

Fig.7 does not depict the change of Bo concentration with time. Therefore, Fig.7 cannot be used to make judgments on the removal rate (kinetics) of the adsorption process. This must be rewritten, and Fig.7 must be replaced by a figure containing 6 curves (one curve for each dose) depicting the decrease of Bo concentration with time.

Line 250: ” 3.2.3. Adsorption kinetics”.

You must provide at this section the following experimental parameters:

(1) volume of boric solution added in the conical flask,

(2) dose of adsorbent, in mass/L.

(3) mixing (oscillation) intensity.

Lines 254-255: ” The concentration of boron in the solution decreased rapidly with the prolongation of adsorption time (Figure 8).”.

Fig.8 does not depict the change of Bo concentration with time. It shows how MNPs-NMDG adsorption capacity increased with time. Therefore, Fig.8 cannot be used to make judgments on how rapid the concentration of boron decreased. This must be rewritten, and Fig.8 must be replaced by a figure containing curves depicting the decrease of Bo concentration with time. If Fig.6 and Fig.7 are changed as I previously suggested, then, in my opinion, Fig.8 can be omitted.

Lines 268-269: ”The pseudo-first-order dynamic equation and pseudo-second-order dynamic equation are shown in formulas (4) and (5):”

You must add a reference here!

Lines 278-281: ”adsorption equilibrium is 4.4852 mg/g, which is quite different from the equilibrium adsorption capacity of 9.2171 mg/g in the experiment. However, the adsorption equilibrium amount of pseudo-second-order kinetics in adsorption equilibrium is 9.5979 mg/g,”

4.48 instead of 4.4852

9.21 instead of 9.2171

9.59 instead of 9.5979

Lines 282-285: ”the adsorption of boron by MNPs-NMDG conforms to the pseudo- second-order kinetic model, that is, the adsorption rate is controlled by chemical adsorption, involving electronic covalent or electronic migration between the adsorbate and adsorbent”

You must add a reference here!

Line 288: ”Table 1. Kinetic parameters of MNPs-NMDG for boron adsorption”

4.48 instead of 4.4852

9.59 instead of 9.5979

0.41 instead of 0.4173

0.08 instead of 0.0822

Line 290: ” 3.2.4. Adsorption Isotherm”.

You must provide at this section the following experimental parameters:

(1) volume of boric solution added in the conical flask,

(2) mixing (oscillation) intensity.

(3) replace ”1 mg adsorbent” with the correspondent dose (mass/L)

Line 300: ” the mathematical linear formula is shown in Formula (6):”

You must add a reference here!

Line 307: ”In this experiment, 1/n is 0.1442”

0.14 instead of 0.1442

Line 309: ” Table 2. Freundlich adsorption isotherm fitting parameters”

0.14 instead of 0.1442

6.02 instead of 6.0295

Line 313: ” 3.2.5. Dynamic adsorption”.

Change ”dynamic method” with ”flow-through column experiments”

You must provide at this section the following experimental parameters:

(1) mass of MNPs-NMDG in the column,

(2) the experimental concentration of boric acid solution.

Lines 320-321: ”Generally, this remarkable leap indicates that the composite material has high selectivity for boron ions in desalinated seawater”

You must add some references here!

Lines 358-359: ”and the static adsorption capacity is 9.2171 mg/g”

9.21 instead of 9.2171

Author Response

(The authors gave the same response as above.)

Reviewer 3 Report

The manuscript entitled “Novel Magnetic Nano-sorbent Functionalized of N-methyl-D- glucamine for Boron Removal from Desalinated Seawater” needs to fulfil the following requirements before it can be accepted for the publication.

  1. The English language needs to be corrected, there are so many grammatical errors, for example, Line 48, “however, although” were used together. In a similar way, there are many errors like that throughout the manuscript, please check.
  2. It is required to abbreviate properly when introducing for the first time, for example, line 75, MNPs were introduced directly without any abbreviation.
  3. In the last paragraph of Introduction section, the term “however” has been used twice, please check.
  4. The authors mentioned in the Introductions section that “However, NMDG has small molecular weight and is easily soluble in water, so it is necessary to develop insoluble NMDG derivatives”, how far the authors succeeded in this direction here?
  5. The lines “In this study, MNPs-NMDG, a novel boron removal adsorbent based on magnetic nanomaterials and boron-specific functional materials, was insoluble in water and magnetic, which was convenient for separation and recovery from water and achieved good boron removal effect” are not clear, totally confusing, please check. What the authors are trying to say by this statement?
  6. The line, “The new adsorbent has great potential to remove boric acid from desalination seawater” is confusing, need to check.
  7. What is the basis/proof that the SiO2 is getting wrapped onto the Fe3O4 nanoparticles? There may be a chance that the SiO2 particles can get formed in the solution separately, or the Fe3O4 is getting wrapped onto the surface of SiO2 particles (like SiO2@Fe3O4)? Except the hypothesis, there is no proper characterization in this direction. The authors need to confirm this first.
  8. Based on the synthesis method that the authors provided, I say that the Fe3O4 and SiO2 particles are just loaded/adsorbed onto the surface of N-methyl-D-glucamine molecules. Which is totally different from the author’s hypothesis, so how the authors defend their view of Fe3O4 wrapped SiO2 bonded NMDG structure?
  9. From the SEM analysis provided in line 171-178, the authors mentioned that the Fe3O4 and SiO2 particles are spherical in shape. How do they confirm that they are spherical, without providing the HRTEM of respective samples. The authors are required to provide the HRTEM images of all the three samples of pure Fe3O4, Fe3O4@SiO2, and Fe3O4@SiO2-NMDG.
  10. Similarly, the FTIR of all three samples of Fe3O4, Fe3O4@SiO2, and Fe3O4@SiO2-NMDG to get clear idea of synthesis. Also, there is proper indication of SiO2 presence in the composite by the FTIR.
  11. From the XRD analysis, the authors are expected to observe a broader peak of NMDG in the range of 20-30 degrees. Also, there is no indication of SiO2 presence in the composite.
  12. Lines 279-285, totally confusing. Also, the word “which” has been used twice within a sentence.
  13. The authors need to compare their results with that of literature studies so as to confirm the efficiency of their study.
  14. The authors are required to check the boron removal studies by taking pure NMDG polymer to see the efficiency. Also, this tells whether the availability of Fe3O4 and SiO2 can make a difference to the adsorption of boron or not?
  15. The porosity, elemental composition, particle size, zeta potential analysis of the composite are strictly required to confirm the mechanism proposed by the authors.    

Author Response

(The authors gave the same response as above.)

Round 2

Reviewer 1 Report

COMMENTS FOR AUTHORS

Manuscript ID: water-1655800

Title:A Novel Magnetic Nano-sorbent Functionalized of N-methyl-D- 2glucamine for Boron Removal from Desalinated Seawater

As a reviewer of your paper I have to thank you for submitting this paper for Water, and advise you that it is acceptable for publication in its present form

Reviewer 2 Report

The manuscript can be accepted for publication in its present form

Reviewer 3 Report

All the comments raised by me are being addressed, some of them with at most satisfaction and some they transferred to the future work. Based on the changes that the authors made, I still accept the work for publication in its present form.